# iPrune: A magnitude based unstructured pruning method for efficient Hardware implementation of Binary Neural Networks

## Abstract

Modern image recognition models span millions of parameters occupying several megabytes and sometimes gigabytes of space, making it difficult to run on resource constrained edge hardware. Binary Neural Networks address this problem by reducing the memory requirements (one single bit per weight and/or activation). The computation requirement and power consumption are also reduced accordingly. Nevertheless, in such networks, each neuron has a large number of inputs, making it difficult to implement them efficiently in binary hardware accelerators, especially LUT-based approaches.

In this work, we present a pruning algorithm and associated results on convolutional and dense layers from aforementioned binary networks. We reduce the computation by 4-70x and the memory by 190-2200x with less than 2% loss of accuracy on MNIST and less than 3% loss of accuracy on CIFAR-10 compared to full precision, fully connected equivalents. Compared to very recent work on pruning for binary networks, we still have a gain of 1% on the precision and up to 30% reduction in memory (526KiB vs 750KiB).

## 1 Introduction

AI has become ubiquitous in our daily lives, be it the use of search engines to find information or automatically tagging photos on social media. While the predictive accuracy of these models have grown, the size and computation requirements have grown by several orders of magnitude or more, making AI a niche field where only few can participate. This practice is also proving to be environmentally unfriendly with Strubell et al. (2019) estimating the footprint of training an NLP model to be 626,000 pounds of carbon dioxide equivalent.

To counter this growing computation need, it is important to develop hardware accelerators which are easily re-configurable. A Field Programmable Gate Array (FPGA) is an integrated circuit designed to be reconfigured by the user and acts as a blank canvas for implementing custom circuits (in our case custom models). FPGAs embed millions of lookup tables (LUTs) that "compute" (actually fetch) the output given the values of their binary inputs. The problem is that these LUTs are limited in size (typically 6-12 inputs) and using multiple LUTs per neuron is less efficient. This raises the need for both binarizing our weights (and/or activation) and reducing the number of inputs per neuron.

In Sections 2 and 3 we go through existing works on binarization and pruning of fully connected, full precision networks. In Section 4, we introduce iPrune, an effective way to prune the weights of binary networks. In Section 5, we compare the accuracy, memory and computation requirements of these network implementations against their full precision, fully connected equivalents. In Section 6, we report some intriguing results about a-priori pruning versus post-training pruning. Section 7 concludes this work.

## 2 Binarization Techniques

Binarization is the process of converting the weights and/or activations to a 1-bit representation ($\pm 1$) instead of the classic N-bit representation (typically 8-bit, 16-bit or 32-bit). This gives us a direct

memory saving of up to 32x. There are several approaches in literature on binarization. Throughout our experiments, we worked with BinaryConnect whose details are given in the next subsection.

## 2.1 BINARYCONNECT

Courbariaux et al. (2015) introduced BinaryConnect, the first paper of its kind, with details on binarization of both convolutional and dense weights. There are two sub-types based on how the binarization is performed.

- Deterministic: The weights are binarized deterministically; specifically, using the sign function on the hard-sigmoid of the weight.

- Stochastic: The weights are set to -1 with probability $(1-\sigma(W_i))$ and 1 with probability $\sigma(W_i)$ where $\sigma$ is the hard-sigmoid function.

The hard-sigmoid function mentioned above is given by:

$$\sigma(x) = clip\left(\frac{x+1}{2}, 0, 1\right) \tag{1}$$

During the back-propagation, the **full precision** weights are updated based on the gradient with respect to the binary weights. The pseudocode for BinaryConnect is given in Algorithm 1

---
**Algorithm 1** Algorithm for BinaryConnect

**1. Forward propagation:**
    $w_b \leftarrow binarize(w_{t-1})$
    **for** $k = 1 : L$ **do**
        compute $a_k$ knowing $a_{k-1}$, $w_b$ and $b_{t-1}$
    **end for**
**2. Backward propagation**
    **for** $k = L : 2 : -1$ **do**
        Compute $\frac{\partial C}{\partial a_{k-1}}$ knowing $\frac{\partial C}{\partial a_k}$ and $w_b$
    **end for**
**3. Parameter update**
    Compute $\frac{\partial C}{\partial w_b}$ and $\frac{\partial C}{\partial b_{t-1}}$ knowing $\frac{\partial C}{\partial a_k}$ and $a_{k-1}$
    $w_t \leftarrow clip(w_{t-1} - \eta\frac{\partial C}{\partial w_b})$
    $b_t \leftarrow b_{t-1} - \eta\frac{\partial C}{\partial b_{t-1}}$

---

## 3 PRUNING TECHNIQUES

Hoefler et al. (2021) discusses in detail different approaches to pruning and was used as a reference for our methods. The different types of pruning techniques are given in Figure 1. A few approaches are described in detail in Sections 3.1 and 3.2

## 3.1 STRUCTURE VS UNSTRUCTURED PRUNING

Structured pruning is the removal of neurons or weights in a structured fashion i.e. based on a fixed pattern. Common methods for structured pruning are neuron pruning (since this is just a row/column removed from the weight matrix) and filter/transformer head removal. Similarly, strided removal of weights after an offset is also an example of structured pruning.

On the other hand, unstructured pruning refers to the pruning method where there are no patterns in how the weights are removed. While this does not help with easier matrix calculation, it tends to provide higher baselines than structured pruning and can often be converted to structured pruning through a minor set of modifications.

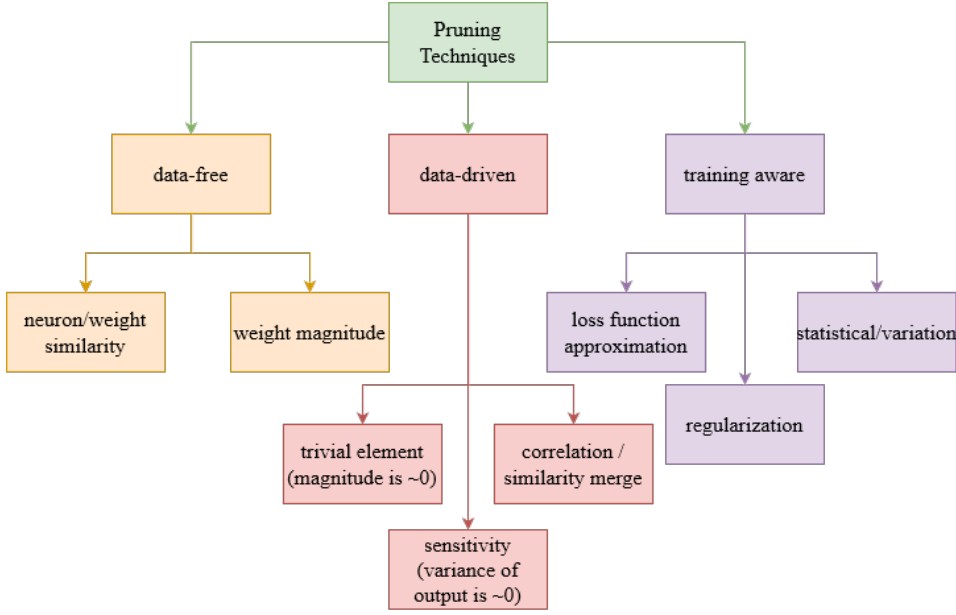

Figure 1: Schemes of pruning described in Hoefler et al. (2021)

## 3.2 MAGNITUDE PRUNING

It is a simple and effective selection scheme where the magnitude of the weights is used as a metric to determine which weights to drop and which weights to keep. After sparsifying the network this way, retraining is done to get high accuracies again.

There are three ways to perform magnitude pruning; globally, layer-wise or neuron-wise. Global magnitude pruning takes all weights from all layers and keeps the top-k weights among them. This often has the problem of vanishing gradients or even vanishing layers (because most weights have been removed from the layer). Figure 2 gives the distribution of weights before and after applying global magnitude pruning.

Layer-wise magnitude pruning (used in Guerra et al. (2020)) keeps the top-k weights across all neurons per layer and overcomes the issue of vanishing layers. This is still difficult to implement in LUTs, because some neurons will have a large number of connections while others will have very few after pruning.

Lastly, neuron-wise magnitude pruning keeps the top-k weights for each neuron. This overcomes both the issues of vanishing layers/gradients and LUT implementation, but often comes at the cost of low initial accuracy which can be solved by retraining the network. In all our experiments, we use neuron-wise magnitude pruning.

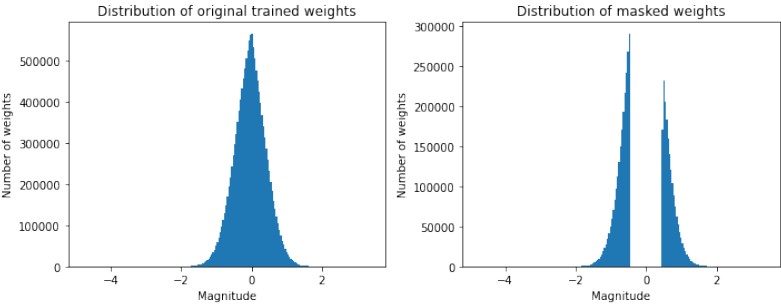

Figure 2: Weight distribution before and after applying global magnitude pruning on custom VGG model for CIFAR-10

### 3.3 LOTTERY TICKET HYPOTHESIS

The Lottery Ticket Hypothesis initially introduced in Frankle & Carbin (2018), and extended to quantized networks in Diffenderfer & Kailkhura (2021) theorizes that there exists optimal subnetworks in the initialization of an overparameterized network. The authors prove theoretically and validate practically this claim in their papers. They introduce a learnable parameter "score" which decide the weights to retain and the ones to drop. Diffenderfer & Kailkhura (2021) claims to reduce network sizes by 50% with negligible reduction or sometimes even increase in accuracy compared to the fully connected counterparts.

## 4 IPRUNE

iPrune is a magnitude based unstructured pruning technique to reduce the number of inputs to each neuron of a particular layer in a neural network.

For dense layers, the magnitude is computed and the top-k weights are chosen for each neuron of the current layers (top-k columns for each row in the weight matrix of shape (out_features, in_features)). For convolutions, the L1 norm across the kernel dimensions (kx, ky) is computed to reduce it to an (out_channels, in_channels) representation. This is then followed by pruning using top-k weights similar to the dense case.

Disconnecting neurons from previous layers can be easily accomplished in most deep-learning frameworks with the use of a mask. Neurons which remain connected have a mask value 1 and the others have 0. In Figure 3, for a given example mask and weight, the computation of the masked weight is shown. The figure also highlights the structure of the mask.

Algorithm 2 provides pseudocode for iPrune. Note that the update step, performed on the full precision weights uses the gradients with respect to the binary masked weights $\left( \frac{\partial C}{\partial w_m} \right)$.

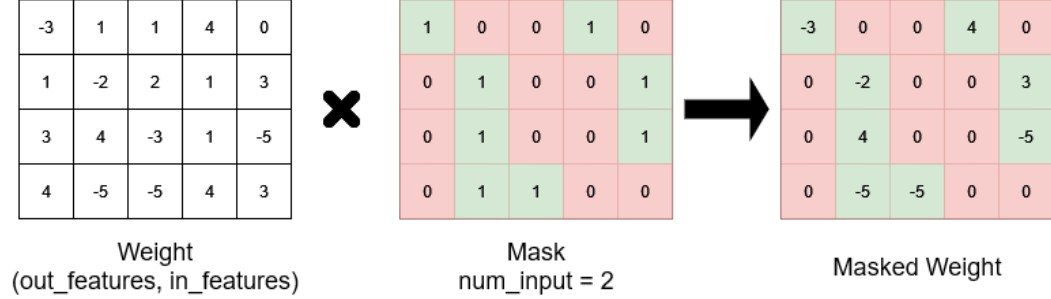

Figure 3: iPrune Masking procedure

## 5 EXPERIMENTAL RESULTS

We performed tests on MNIST (LeCun et al. (1998)) and CIFAR-10 (Krizhevsky & Hinton (2009)) datasets. For all the results given below, the network architecture used for MNIST was (1024D - 1024D - 10D) and for CIFAR-10 a modified version of VGG-Small, specifically: (128C3 - 128C3 - MP2 - 256C3 - 256C3 - MP2 - 512C3 - 512C3 - MP2 - 1024D - 1024D - 10D)

For MNIST, no preprocessing was performed. We skipped the last layer while pruning. For CIFAR-10 we applied ZCA preprocessing as was suggested in Courbariaux et al. (2015) for gains of around 2-3% in accuracy. We applied weight-decay (5e-2) on the last layer (dense) of the network and used the square hinge loss as the loss function. This together behaves like an L2-SVM block. We use the Adam optimizer with initial learning rate of 3e-3 with no learning rate scheduler. While pruning, we skipped the first two convolution layers and the last dense layer.

Figure 4 shows the effect of iPrune on the weight values for the models we trained. Notice how the number of neurons with weight magnitude close to zero reduces in both cases. At the same time the

---

**Algorithm 2** Algorithm for iPrune

**0. Train a fully connected model**
**1. Compute the mask - Done only once**
  $w.mask = \text{find\_mask}(w)$
**2. Forward propagation:**
 $w_b \leftarrow binarize(w)$
 $w_m = w_b * w.mask$
 **for** $k = 1 : L$ **do**
  compute $a_k$ knowing $a_{k-1}$, $w_m$ and $b$
 **end for**
**3. Backward propagation**
 **for** $k = L : 2 : -1$ **do**
  Compute $\frac{\partial C}{\partial a_{k-1}}$ knowing $\frac{\partial C}{\partial a_k}$ and $w_m$
 **end for**
**4. Parameter update**
 Compute $\frac{\partial C}{\partial w_m}$ and $\frac{\partial C}{\partial b}$ knowing $\frac{\partial C}{\partial a_k}$, $a_{k-1}$ and $w.mask$
 $w \leftarrow clip(w - \eta \frac{\partial C}{\partial w_m})$
 $b \leftarrow b - \eta \frac{\partial C}{\partial b}$

---

number of these weights is not zero as we are applying neuron wise pruning, not global pruning. Also note that in the BinaryConnect case, the original distribution is a mixture of three Gaussians centered at -1, +1 and 0 while in the FP case, there is only one Gaussian centered at 0.

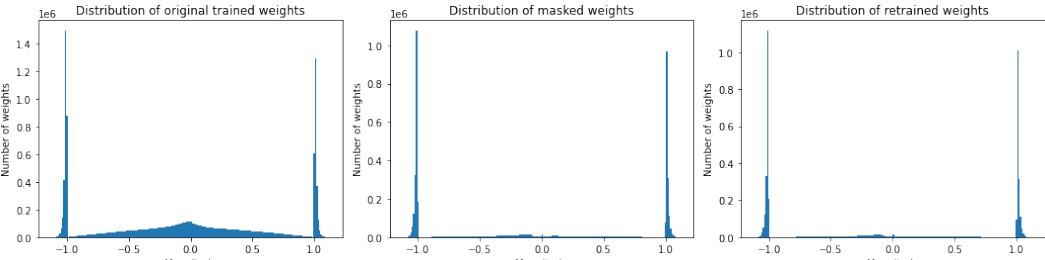

(a) BinaryConnect model with 30% weights remaining in all layers except first two convolutional and last dense

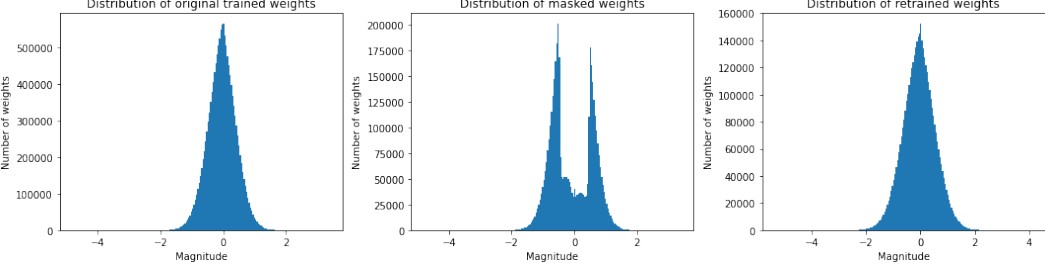

(b) Full Precision model with 30% weights remaining in all layers except first two convolutional and last dense

Figure 4: Weight distribution before, after pruning and after retraining a custom VGG model

Table 1 gives the results after applying iPrune on MNIST. There is less than 2.3% difference between the fully connected, full precision network and pruned BinaryConnect deterministic model with 7/8 inputs. The memory gain is around **2200x** and computation gain is around **70x**.

Table 2 gives the accuracy memory and computation requirement for different percentages of weights remaining per neuron. It is clear that we can reduce the memory by 1.9x-4.7x (remaining weights 50%-20%) and computation by 1.9x-4.4x, compared to the fully connected BinaryConnect deterministic network, with negligible loss in accuracy (less than 1%).

Table 1: Accuracy on pruning models for MNIST with "Number of inputs" connections to each neuron of current layer to neurons from previous layer.

| Method | Baseline (1024 inputs) | Number of inputs | Accuracy |
|---|---|---|---|
| BinaryConnect Deterministic | 98.08% | 5 | 94.47% |
| | | 6 | 95.19% |
| | | 7 | 95.98% |
| | | 8 | 96.01% |
| Floating Point | 98.27% | 5 | 97.51% |
| | | 6 | 97.80% |
| | | 7 | 97.71% |
| | | 8 | 98.06% |

Table 2: Accuracy, memory and computation requirements with different percentages of weight remaining for each neuron while pruning models for CIFAR-10.

| Percentage weights remaining | Accuracy | Memory | Computation (Ops) |
|---|---|---|---|
| 1.0 % | 72.89 % | 63.787KiB | 4.30361e+08 |
| 5.0 % | 77.09 % | 103.047KiB | 3.72070e+08 |
| 10.0 % | 83.41 % | 187.953KiB | 6.01394e+08 |
| 20.0 % | 86.20 % | 357.359KiB | 1.05768e+09 |
| 30.0 % | 87.54 % | 526.047KiB | 1.50689e+09 |
| 50.0 % | 87.71 % | 865.672KiB | 2.42419e+09 |
| 80.0 % | 87.86 % | 1.340MiB | 3.77653e+09 |
| 100.0 % | 87.64 % | 1.672MiB | 4.69383e+09 |

Since we observed in our early experiments with MNIST that dense layers can be pruned further, we performed a gridsearch with different percentages/number of weights remaining in convolution and dense layers. The results are given in Table 3. The baseline accuracy for these models is 87.64%. The results mostly show a trend of increasing accuracy with increasing number of dense/percentage of convolution weights per neuron apart from a few anomalies. We can also observe that the computation and memory requirements are highly skewed towards reduction in convolution parameters than dense parameters.

Also note, based on Table 2, Table 3, and our baseline accuracy for fully connected, full precision model (88.21%), that with less than 3% reduction in accuracy, we were able to save up to **180x** in memory and up to **4.4x** in computation.

Previous work on this topic (Guerra et al. (2020)), which used layer wise unstructured pruning, was able to prune BinaryConnect models to 750 KB size with 1.4x reduction in computation while our models with **1% higher accuracy** can be pruned to as low as **526 KB** with upto **3.1x reduction** in computation. Compared to pruning full precision models from Li et al. (2016) we reduce memory by upto 41x (526KB vs 21.6 MB) with 6% reduction in accuracy (Note that the comparisons have been done with results of closest equivalent VGG models reported by the authors).

## 6 IPRUNE FOR RANDOMLY INITIALIZED NETWORKS

As an ablation study, we compared iPrune on a fully trained network for MNIST against iPrune on a randomly initialized network. Table 5 shows the results of iPrune on a randomly initialized network vs a fully trained network.

The idea behind this is that the initialization of networks greatly affects the magnitude of the gradient and subsequently the final value after training. Existing work on such "Lottery Tickets" have been discussed in Section 3. Those methods use a learnable metric "score" to determine the mask. This score is updated while training. Conversely, in this ablation, we compute the mask once, just after initialization, and assume that this is the "Lottery Ticket" network. We allow the update of weights while training to allow the model to fit better to the data instead of updating the mask itself.

Table 3: Accuracy, Memory and computation requirements of different combinations of percentage convolution weights remaining per neuron and number of dense weights remaining per neuron for iPrune for CIFAR-10 BinaryConnect Deterministic.

| Dense \ Conv | 8% | 20% | 30% | 50 % |
|---|---|---|---|---|
| 7 | 78.11% | 80.93% | 82.43% | 83.65% |
| | 63.569KiB | 128.819KiB | 182.257KiB | 291.382KiB |
| | 5.08453e+08 | 1.05581e+09 | 1.50408e+09 | 2.41948e+09 |
| 8 | 78.33% | 81.21% | 82.92% | 84.30% |
| | 63.826KiB | 129.076KiB | 182.514KiB | 291.639KiB |
| | 5.08455e+08 | 1.05581e+09 | 1.50408e+09 | 2.41948e+09 |
| 10 | 78.40% | 81.68% | 83.01% | 85.10% |
| | 64.344KiB | 129.594KiB | 183.031KiB | 292.156KiB |
| | 5.08459e+08 | 1.05582e+09 | 1.50408e+09 | 2.41949e+09 |
| 20 | 78.84% | 82.07% | 83.90% | 85.49% |
| | 66.852KiB | 132.102KiB | 185.540KiB | 294.665KiB |
| | 5.08479e+08 | 1.05584e+09 | 1.50410e+09 | 2.41951e+09 |
| 50 | 78.60% | 82.33% | 83.49% | 84.79% |
| | 74.359KiB | 139.609KiB | 193.047KiB | 302.172KiB |
| | 5.08541e+08 | 1.05590e+09 | 1.50416e+09 | 2.41957e+09 |
| 100 | 79.05% | 83.29% | 84.83% | 85.32% |
| | 86.859KiB | 152.109KiB | 205.547KiB | 314.672KiB |
| | 5.08643e+08 | 1.05600e+09 | 1.50427e+09 | 2.41967e+09 |

Table 4: Comparison of our results with previous works, Guerra et al. (2020) on binary pruning and Li et al. (2016) on full precision networks. The source for our results is in Table 2

| Paper | Model | Baseline | Pruned | Memory | Memory Reduction | Computation Reduction |
|---|---|---|---|---|---|---|
| Li et al. (2016) | VGG-16 (Full Precision) | 93.25% | 93.40% | 21.6MiB | $\frac{1}{28.8}$x | 1.5x |
| Guerra et al. (2020) | VGG-11 (Binary Connect) | 87.60% | 86.53% | 750KiB | 1x | 1.4x |
| Ours | Custom VGG (30% pruned) | 87.64% | 87.54% | 526KiB | 1.4x | 3.1x |
| | Custom VGG (20% pruned) | 87.64% | 86.20% | 357KiB | 2.0x | 4.4x |

Table 5: Comparison between trained, pruned and retrained network and a randomly initialized, pruned and trained BinaryConnect Deterministic network. The baseline accuracy for the Train-Prune-Train case is 87.64%.

| Number of inputs | Train-Prune-Train | Prune-Train |
|---|---|---|
| 20% | 86.20% | 85.83% |
| 30% | 87.54% | 86.13% |
| 50% | 87.71% | 86.52% |
| 80% | 87.86% | 86.60% |

The results show that a randomly initialized network can be pruned to less than 1.5% reduction in accuracy for the same training parameters. This encourages further investigation in future works on larger models to validate our hypothesis. If this phenomenon is always observed, it can greatly help reduce the computation cost by removing the need to train larger models initially.

## 7 CONCLUSION AND FUTURE WORK

We were able to demonstrate pruning for binary neural networks and reduce the memory and computation requirement by more than 2200x and 70x on MNIST and 190x and 4.4x on CIFAR-10, respectively, compared to full precision, fully connected equivalents. Our CIFAR-10 results had 1% increase in accuracy but was only 70% of the size of the model from Guerra et al. (2020) with a reduction in computation time by 3.1x vs 1.4x in that paper.

The ability to prune randomly initialized neural networks and train to reasonably high accuracies is an interesting outcome and is worth investigating. Finally, extension of these algorithms for other forms of quantization (logarithmic, k-bit, etc.) would also be an interesting avenue for research.

We are also working on porting the trained weights to an FPGA to observe the gain in energy requirements and inference time. With such low number of inputs (fan in), we expect to be able to fit our neurons in lookup tables to further boost the performance.

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
