# OpenReview forum: "iPrune: A Magnitude Based Unstructured Pruning Method for Efficient Binary Networks in Hardware"
_ICLR.cc/2022/Conference — ICLR 2022 Submitted_

### Official Review · Reviewer_TbnU · 2021-10-30

**Correctness:** 2
**Technical Novelty And Significance:** 1
**Empirical Novelty And Significance:** Not applicable
**Recommendation:** 3
**Confidence:** 5

**Main Review:**

The authors are trying to address an important task in deep learning which is neural network compression.

However, the authors introduced too much background of the network binarization and unstructured pruning, while the discussion about the proposed method is not enough.

The proposed methods are built heavily on the previous methods, it would be better if the authors could propose more novel ideas.

Many technique details are not clear. For example, in Fig.3. the output of the masked weights are integers, which is confusing because the authors claimed that this paper is working on pruning the binary neural networks.


**Summary Of The Paper:**

This paper introduces a neural network compression method built on magnitude-based unstructured pruning and binarization techniques.

**Summary Of The Review:**

In general, I think this paper is not ready for publish and still has a big room for improvement in terms of novelty, writing, and technique contributions.

---

### Official Review · Reviewer_r65C · 2021-11-01

**Correctness:** 4
**Technical Novelty And Significance:** 1
**Empirical Novelty And Significance:** 1
**Recommendation:** 3
**Confidence:** 5

**Main Review:**

Strengths:
1) I think the main strength of this paper is successful exploitation of two existing method to reduce computational complexity and memory requirements of small datasets.
2) Extensive simulation results and investigating different aspects of the proposed method is another strength of this paper.

Weaknesses:
1) The contributions of this paper are not novel. The proposed method is a combination of two existing methods and it overlaps with a previously-published work (please see https://openreview.net/pdf?id=r1fYuytex).
2) It was tested on small datasets only. It would be great if the same level of memory and computation reduction can be achieved on larger datasets such as ImageNet.

**Summary Of The Paper:**

This paper combines two existing methods (i.e., binarization and pruning) to reduce both computational complexity and memory requirements of deep neural networks (DNNs). Binarryconnect method was exploited to binarize DNNs whereas neuron-wise magnitude pruning was used to remove unnecessary input activations. It was then shown that the combination of these two methods can significantly reduce both memory and computation on CIFAR-10 and MNIST datasets.

**Summary Of The Review:**

As mentioned in the paper, this work relies on two existing works. Even though successful adaptation of existing works is challenging and valuable, I believe the contribution of this paper is still very limited since it overlaps with an existing work (i.e., https://openreview.net/pdf?id=r1fYuytex).

---

### Official Review · Reviewer_1172 · 2021-11-11

**Correctness:** 2
**Technical Novelty And Significance:** 2
**Empirical Novelty And Significance:** Not applicable
**Recommendation:** 3
**Confidence:** 4

**Main Review:**

The main strengths of the paper are:

1) Review and discussion of current approaches that the paper is building on.
2) The paper presents a good amount of experimental results.

However, they are both very limited in scope:

1) The paper only presents very limited approaches on network binarization and pruning - which is a very hot topic, and a ton of research has been done in recent years.
2) The authors focus on two very modest (easy) datasets, which don't require big models, and on which pruning/binarizations techniques are expected to work.

Additionally:

3) The paper uses very limited amount of page real-estate to describe the iPrune approach, and immediately jumps to experimental results.
4) It is not clear what the authors claim as novelty.
5) The authors claim 190-2200x memory reduction for MNIST, which seems unsubstantiated in the paper.

**Summary Of The Paper:**

The paper introduces an algorithm for pruning binary networks. The authors combine existing binarization approaches (binary connect) and use magnitude based unstructured pruning of the binarized network.

**Summary Of The Review:**

Please see above.

In order for this work to be acceptance quality, the authors should at least provide more details about the iPrune approach and compare it with more state of the art approaches. Additionally, the authors should perform experiments on more challenging datasets in order to substantiate their claims.

---

### Official Review · Reviewer_MtDD · 2021-11-12

**Correctness:** 2
**Technical Novelty And Significance:** 2
**Empirical Novelty And Significance:** 1
**Recommendation:** 3
**Confidence:** 3

**Main Review:**

Strength:
- This paper suggested a new method of pruning binary neural networks.

Weaknesses:
- The paper writing is sloppy.
    - Algorithms 1 and 2 use notations that are not defined. What are a_k, b, w_k, t? What are find_mask() doing? What is the range of clip function?
    - The main part is Algorithm 2, and needs better description at the main body. The author should not let the readers to understand on their own after looking at Algorithm 2.
    - Notation is vague. In algorithm 1, w_b is binary weight, and w_t seems to be the weight at step t. We should distinguish them instead of using the same notation of under-script.
- I cannot get the novelty of this work.
    - Where can we check that memory gain is around 2200x and computation gain is around 70x? The authors claim this after mentioning Table 1, but cannot find how Table 1 implies this result.
    - In Table 4, the gain is not significant.
- Need better comparison with other schemes
    - Pruning binary/non-binary neural networks have been considered in various literatures. It is bit surprise that the authors only cited less than 10 papers.
    - The authors only compared the performance with 2 papers, but I think there should be a large number of pruning techniques for binary neural networks.


**Summary Of The Paper:**

This paper suggests iPrune, a magnitude based unstructured pruning technique which reduces the number of inputs to each neuron. This scheme reduces the memory, computation, and power consumption for training binary neural networks.


**Summary Of The Review:**

This paper needs better illustration of their algorithm and better justification of the novelty.

---

### Decision · Program_Chairs · 2022-01-20

**Decision:**

Reject

**Comment:**

This paper deals with a problem of significant practical relevance: memory efficient neural networks. The authors propose some pruning methods for binary networks. However, several weaknesses were identified by the reviewers (novelty, lack of extensive experiments, problems with the presentation of the paper), and several valid points of concern were raised. These points of criticism were not adequately addressed, hence the paper in its current form cannot be recommended for publication.